# Dynamic Functional Connectivity of Emotion Processing in Beta Band with Naturalistic Emotion Stimuli

**DOI:** 10.3390/brainsci12081106

**Published:** 2022-08-19

**Authors:** Sudhakar Mishra, Narayanan Srinivasan, Uma Shanker Tiwary

**Affiliations:** 1Indian Institute of Information Technology Allahabad, Prayagraj 211012, India; 2Indian Institute of Technology Kanpur, Kanpur 208016, India

**Keywords:** naturalistic study, dynamic functional connectivity, phase-locking value, arousal, dominance

## Abstract

While naturalistic stimuli, such as movies, better represent the complexity of the real world and are perhaps crucial to understanding the dynamics of emotion processing, there is limited research on emotions with naturalistic stimuli. There is a need to understand the temporal dynamics of emotion processing and their relationship to different dimensions of emotion experience. In addition, there is a need to understand the dynamics of functional connectivity underlying different emotional experiences that occur during or prior to such experiences. To address these questions, we recorded the EEG of participants and asked them to mark the temporal location of their emotional experience as they watched a video. We also obtained self-assessment ratings for emotional multimedia stimuli. We calculated dynamic functional the connectivity (DFC) patterns in all the frequency bands, including information about hubs in the network. The change in functional networks was quantified in terms of temporal variability, which was then used in regression analysis to evaluate whether temporal variability in DFC (tvDFC) could predict different dimensions of emotional experience. We observed that the connectivity patterns in the upper beta band could differentiate emotion categories better during or prior to the reported emotional experience. The temporal variability in functional connectivity dynamics is primarily related to emotional arousal followed by dominance. The hubs in the functional networks were found across the right frontal and bilateral parietal lobes, which have been reported to facilitate affect, interoception, action, and memory-related processing. Since our study was performed with naturalistic real-life resembling emotional videos, the study contributes significantly to understanding the dynamics of emotion processing. The results support constructivist theories of emotional experience and show that changes in dynamic functional connectivity can predict aspects of our emotional experience.

## 1. Introduction

Real-life emotional experiences are dynamic and contextual. The lack of context may result in ambiguity, which is a problem with isolated stimuli such as pictures (e.g., Ekman’s face [1]) and sounds (e.g., oddball tones [2]). In contrast, emotional films provide a contextual and narrative structure for eliciting a more representative and better emotional experience [3,4]. In addition, they may preserve the natural timing relations between the constituting functional components, which could be studied using EEG.

To date, EEG has been used to understand emotion processing with mostly static [5,6,7] and a few dynamic stimuli [8,9,10,11]. More beta oscillatory activity in frontal, central and parietal sites were found with affective pictures [6]. International Affective Picture System (IAPS) images with negative valence induced more beta power than neutral images when participants were explicitly instructed to rate the strength of the picture’s influence on their own emotional state [7]. Significant enhancement in the power of the beta band (in the range 25–30 Hz) was observed for negative pictures in comparison to neutral pictures [12]. There is increased long-distance EEG connectivity in the parietal lobe and temporal lobe in the beta band during affective picture processing [9]. A high synchronous activity between the posterior parietal electrodes and the prefrontal electrodes in the beta band was observed with highly arousing affective images [5]. In addition to beta oscillations, brain activity in other frequency bands has also been reported for affective images [7,13,14,15]. For instance, a study found higher power for pleasant and unpleasant emotional pictures than neutral pictures in the delta, theta, and alpha bands [13]. Other than beta event-related synchronization (ERS), an higher elicited delta ERS response was observed with positive and negative pictures than with neutral pictures [7]. For moderate and high arousal IAPS pictures, larger synchronization in the theta band over bilateral posterior regions was observed [14]. In addition, larger ERS activity in the alpha band (alpha divided into three bands) over the occipital and posterior regions was also observed [14].

Even though multimodal film stimuli should be more effective in eliciting emotions [3], EEG studies with such stimuli are very limited. Moreover, they did not probe felt emotions or limited themselves to very few emotional categories [8]. For instance, multimodal emotion perception (not felt emotions) for anger and fear for complex stimuli such as short video clips, body language, and audio–visual information reported early interaction of audio-visual modalities in the alpha and beta bands [8]. In another study [9], participants were shown emotional movies eliciting positive and negative emotions and parietal and temporal lobe electrodes had higher accuracy for emotion categorization in the beta band. The inter-regional synchronization of oscillatory activity in the beta band has been shown to be reduced due to emotionally arousing video stimuli [15]. The research using static and multimodal stimuli indicates a complex interplay of frequency bands and emotion processing, which requires further clarification.

Although some studies were performed with dynamic emotional stimuli [8,9,16], they did not explicitly probe brain activity just before an emotional experience. These studies with film stimuli assumed that the emotion would be felt for the whole duration of the film and did not temporally localize emotion to understand the dynamics of processes leading to emotional experience. Analyses based on the whole duration [10,11] may include significant non-emotional processing or mixed processing associated with emotions elicited at different points in time. Given that different emotions can potentially be elicited at different time points with a video, a design with a temporal marker for the emotional experience is needed to understand the mechanisms that underlie the generation of an emotional experience.

Recent studies investigated the functional connectivity of the human brain during rest and emotional conditions [17]. For instance, participants were shown all neutral or all negative IAPS pictures in the static condition and images from different categories were shown between the first and last two images in the dynamic condition [18]. The functional connectivity between prefrontal and parietal scalp electrodes was higher in the beta band in the static condition in comparison to the dynamic condition. Functional connectivity has been rarely studied using multimodal emotional stimuli [19]. Brain networks were investigated using phase-locking-value (PLV) during positive and negative emotions, and they found that PLV values between left posterior (P7) and temporal (T7) locations and between right temporal (T8) and inferior frontal (F8) were significantly different for high and low valenced audio + visual stimuli in the beta band [19].

The temporal variability in the connectivity profile provides information about the task-related re-organization of functional networks and hubs over time that promotes flexibility and adaptability of the networks to dynamic contexts present in naturalistic stimuli. The dynamic functional connectivity (DFC) calculation was proposed to detect the time-varying changes in FC measurements [20]. The most commonly used DFC estimation method is the sliding window approach with fixed-length segments. The network characteristics for each time segment is used for the quantification of temporal variability in the dynamic functional connectivity [21]. Calculation of temporal variability of DFC (tvDFC) is a relatively new developed analysis technique [22,23], which was used in task-free [24], task-based [25,26], and clinical [27,28] conditions.

With respect to emotions, the correlation of dynamics of cohesion within and between salience, default mode (DMN), and executive control networks with the affective dimensions was probed using fMRI while participants watched stressful naturalistic stimuli during the first viewing [29]. During the second viewing, participants recalled and rated their emotional arousal on the arousal scale, followed by continuous ratings of valence values during the third viewing. The dynamic cohesion within the salience network was correlated with emotional arousal. The cohesion between the salience and executive control network was highest during the modest stress-related arousal. The results indicate that the dynamic activity within the functional network is related to emotional arousal. Another study [30] found that the dynamics of functional connectivity covaried with the emotional intensity. With higher emotional intensity, the connectivity between the salience and medial amygdala networks was stronger, supporting constructionist theories of emotion. Continuous enjoyment ratings were associated with the momentary intersubject synchronization in auditory, default mode, and striatal networks, whereas sadness ratings were associated with the limbic and striatal networks [31]. These studies [29,30,31] presented a stimulus more than once to obtain emotional ratings. On the other hand, Lettieri et al. [32] studied the association between time-varying inter-subject brain synchronization (tISFC) to variations in the perceived affective states during movie watching with two different groups. They obtained continuous behavioral ratings from one group and fMRI recordings from another group. The correlation between emotional intensity and polarity was observed with the connectivity dynamics of the default mode and control networks.

The studies described above were performed with fMRI, which has low temporal resolution. In addition, the length of the window to calculate the dynamics of neural synchronization is longer (≈20 s), which may not be suitable for use with the transient or fast-changing nature of the emotions. Moreover, they required participants to watch a stimulus more than once to obtain neural and behavioral data. To date, no EEG study investigated the relationship between simultaneously recorded subjective feedback of felt emotional events embedded within a context and time-varying reorganization of brain activity. EEG research with multimodal emotional stimuli investigating different frequency bands and functional connectivity is very limited and inconclusive. In addition, the temporal dynamics of emotional experience are also not well understood due to the unavailability of information about the time at which such an experience occurs. Furthermore, the knowledge of tvDFC for emotion processing is lacking. Hence, we probed the spatio-temporal dynamics of the emotional experience embedded within a dynamic context and multimedia stimuli. We particularly analyzed temporally marked EEG signals related to emotional events when participants watched film stimuli. Given the importance of the beta band during emotion processing, we expected that the emotion-related functional networks could be more distinguishable in the beta band than in other bands, and the variability in functional networks could be associated with different dimensions of emotional experience.

## 2. Materials and Methods

### 2.1. Participants

Forty-three students participated in order to fulfill a course requirement and signed a written consent form as required by the Institutional Review Board. After removing three participants due to excessive movement, 40 participants were finally considered (Mean age = 23.3, SD = 1.4, female = 3).

### 2.2. Stimuli and Apparatus

The emotional stimuli were taken from the stimuli dataset [33]. The list of emotional stimuli used in the training and EEG experiment is available online at [34].

Non-emotional stimuli were selected from a separate validation study. Twenty participants (different from 40 EEG participants with the same age range) were shown 10 non-emotional stimuli downloaded from YouTube with scenes such as a running train, news item reading, etc. Those stimuli were regarded as non-emotional, which had mean valence and arousal ratings of around 5. Additionally, we checked whether participants rated the stimulus as non-emotional (by not assigning any emotional category to the stimulus). The purpose was to provide a more neutral non-emotional stimulus to avoid accumulation of emotional effects over the course of the experiment. The first non-emotional stimulus was about the world’s longest road routes and the second was an animated history of the Babylonian era.

The videos were presented on a 15.6-inch monitor with a resolution of 800 × 600. The audio was presented using a Sennheiser CX 180 Street II in-Ear Headphone. Responses were obtained using a mouse and keyboard.

### 2.3. Experiment Paradigm

Before starting the main experiment, participants were trained using four emotional stimuli. The training stimuli were also taken from the same dataset, which we validated in an earlier study [35]. During the training, participants were informed about the experiment procedure, rating scales (given a quiz to assure they understood the self-assessment scales well), and trained to perform a mouse click when they felt any emotion and labeling these clicks with emotion categories. We gave participants definitions of the emotional categories from the Oxford Lexico Dictionary [36]. We also asked participants if they were able to understand the categories, and they had no issues understanding these words related to common emotions.

During the main experiment, after reading the instruction screen, the baseline signal (eyes open looking at the cross mark) was recorded for 80 s. Then, 11 one-minute multimedia stimuli (two non-emotional and nine emotional videos) were presented. One non-emotional video was presented just after the baseline recording, whereas another non-emotional video and nine emotional videos, selected randomly out of 16 emotional videos, were presented in random order. The second non-emotional video was presented at some point between the fifth and eighth videos. Every stimulus video was preceded by an inter-stimulus interval and followed by response windows to respond on self-assessment scales: valence, arousal, dominance, liking, familiarity, relevance, and emotional category selection. Valence, arousal and dominance scales ranged, respectively, from unpleasant (1) to pleasant (9); inactive (1) to active (9); and submissive (1) to dominant (9). Liking, familiarity, and relevance scales ranged, respectively, from least (1) to fair (3) to much (5); less-familiarity (1) to high-familiarity (5); and from not related (1) to completely related (5). Emotional categories were presented in the form of a drop-down menu. The complete paradigm is shown in Figure 1. Particularly, physiological arousal is the activation state of emotion, which ranges from inactive to active state Russell and Mehrabian [37]. On the other hand, dominance ranges from feelings of total lack of control or influence on events and surroundings to the opposite extreme of feeling influential and in control [37].

While watching the video, participants marked the moment of their emotional feelings with a mouse click. They could click any number of times while watching the video. After they finished watching the video, they were asked to give an emotion label that best described their felt emotion when they made their clicks. To help recall the felt emotion, participants were shown three frames extracted around the time duration when they did a mouse click for feeling an emotion. This simple method allowed us to localize a participant’s emotional experience temporally while at the same time not interrupting the watching of the video. To provide an emotion category for each click, the participants were given a list of emotions in the form of a drop-down menu for each of the four quadrants of valence-arousal space to freely select the emotion they felt. However, to avoid overlapping with adjacent clicks, participants were explicitly instructed that they should not click again and again for the emotional feeling elicited by a similar event in the scene. Data for 420 emotional events for 24 self-reported emotions were collected from 40 participants.

### 2.4. EEG Data Acquisition and Pre-Processing

EEG recordings were obtained using a 128-channel (Geodesic EEG 400) system with active electrodes. Impedance was below 30 kΩ (consistent with previous evidence [38,39,40,41]). Netstation software was used to acquire the raw signal at 250 Hz sampling rate. The raw signal referenced to the Cz electrode was imported. A Butterworth bandpass filter with the passband 1 Hz to 40 Hz was used. Then the Emotional and baseline events were extracted from the filtered signal with the time duration, respectively, 6 s before clicking to 1 s after the click and 10 s to 70 s. However, in the processing, only seven second long segments were randomly extracted from the baseline signal. We observed that there were only 18 cases of overlapping clicks considering emotional events with a length of seven seconds. We did not consider these 18 clicks.

Each participant’s emotional events were concatenated and checked manually for electrodes with very high amplitude (probably due to the detachment of electrodes due to neck movement [42]). Further, ICA is applied to remove the artifacts, including eye blink, muscle activity, heart activity, line noise, and channel noise. The IC components obtained from ICA were labeled using ICLabel tool [43]. The ICA components were calculated separately for each subject and the number of selected ICA components labeled as “brain signals” by ICLabel tool varied from 25 to 45 across subjects (agreeing with the emotion analysis done by Hsu et al. [44] on dense EEG with 128 channels). The figure depicting EEG preprocessing on the studied data is available (See Figure 2 in [45]).

### 2.5. Analysis

#### 2.5.1. Emotion Grouping

Since we had only a small number of instances per emotion, we grouped 24 self-reported emotions based on two criteria. First, the distance-based proximity of mean (calculated for each emotion) on V-A space, and second, approximately equal instances across emotion groups. The distance-based proximity here is related to the calculation of Euclidean distance among emotional categories. The mean valence-arousal vector was used to calculate the Euclidean distance. This procedure was earlier adapted by [46]. Based on these criteria, we obtained eight emotion groups with approximately 50 samples in each emotion group (as shown in Figure 2). The mean (sd) ratings of self-assessment scales for each group are presented (see Figure 2). We used this grouping for further analysis.

#### 2.5.2. Frequency Bands

Frequency bands evaluated in the study are as follows. 1–4 Hz (delta), 4–8 Hz (theta), 8–12 Hz (alpha), 12–20 Hz (Lower beta), 20–30 Hz (Upper beta), 30–40 Hz (Gamma).

#### 2.5.3. EEG Signal Segmentation

The duration of the extracted EEG signal corresponding to each emotional event was 7 s. The sampling rate was 250 Hz. Each emotional event signal was divided into segments containing 250 samples (75 samples overlapping). The segmenting was based on the recent work by Lettieri et al. [32]. Like them, we also considered a 33% overlap between consecutive windows. In this way, we obtained nine segments to perform further analyses. Segments are: seg-0 (1–250); seg-1 (175–425); seg-2 (350–600); seg-3 (525–775); seg-4 (700–950); seg-5 (875–1125); seg-6 (1050–1300); seg-7 (1225–1475); seg-8 (1400–1650).

#### 2.5.4. EEG Dynamic Functional Connectivity and Temporal Variability of the Functional Connectivity Profile

We used PLV as a measure for calculating connectivity between EEG Electrode pairs [47]. The obtained connectivity matrix for each emotion group was contrasted with the baseline state connectivity matrix, and a non-parametric approximate permutation test (multiple hypothesis comparisons using a single threshold technique) was calculated. This experimental contrast method helped us deal with the issue of crosstalk to some extent [48]. We assumed that the spurious phase synchronization due to crosstalk would affect both the conditions approximately similarly [49]. However, this method has a limitation as the effects of crosstalk are highly dependent on the amplitude of underlying (noise) sources. Channel pairs were approved to be significant in the permutation test if the Bonferroni-holm adjusted *p*-value is less than the critical α level 0.01. The significant connections are plotted on the head cartoon in Figure 3.

We performed a multiple-comparison non-parametric permutation hypothesis test for related data corrected for family-wise type-1 error (FWER) to find out the significant functional connection among pair of electrodes [50]. The null hypothesis is that permuting the labels of related samples assigned to different experimental conditions leads to an equally likely statistic of interest. That means the statistic for actual labeling is in the confidence interval of obtained permutation distribution after a sufficient number of relabeling. Hence, the *p*-value is the proportion of the statistics greater than or equal to the statistics *t* corresponding to the correctly labeled data. This multiple-comparison test was performed for all electrode pairs. The family-wise type-1 error rate, due to family-wise multiple comparisons, was controlled using a single threshold test. The key idea behind the single threshold test is that given a critical threshold (single threshold), functional connections with statistic values exceeding this threshold have their null hypothesis rejected. The critical threshold was decided using the distribution of the maximum statistics. This distribution of the maximal statistic was computed from the statistic image obtained after several permuted relabeling. The omnibus hypothesis at level α is rejected if the maximal statistic calculated for actual labeling was in the confidence interval 100α% of permutation distribution. The critical value is the (c+1)th largest member of the permutation distribution, where c=⌊αN⌋.

As mentioned above, by definition, the single threshold technique (with maximum statistics) assumes the uniformity of null sampling distribution over all electrode pairs (pairs of electrodes making a link). However, using mean difference statistics may not guarantee this uniformity because an area with high variability in mean difference (across samples) can dominate the maximum statistics. Hence, the statistics should be normalized by this variability in mean difference. To this aim, instead of using mean difference as the statistic, we used t-statistics, which also account for the variability in values across samples. A sample here means an emotional event reported by the subjects. During each permutation (using condition relabeling), the t-statistic was corrected for FWER by considering the maximum statistics of the statistical image. The single threshold technique is claimed to be very effective in FWER [51,52]. Hence, the method we adapted inherently accounts for the variability across subjects.

Calculating distance between connectivity patterns: We obtained the connectivity network representation for each emotion group on the scalp space (Figure 3). For different phases of cooperating and competitive control processes, the subgraphs in functional networks fluctuate [53]. Similarly, for different emotional experiences, there could be fluctuations in subgraphs in functional networks. Following this assumption, a multi-hot embedding vector for each time segment was created from the functional connections. The Euclidean distances between the connectivity vectors for all combinations of emotion pairs at each frequency band and segment were calculated. We used these distances between connectivity networks to find the frequency bands with relatively more information (more information means connectivity distances are higher). The higher the distance between connectivity vectors of pair of emotion groups, the more separation between networks. We calculated the average distance for each emotion pair by pooling the distances across all the segments in each frequency band, giving us 28 average distances at each frequency band. These average distances were compared with the population mean, which was calculated by further taking the average across all the frequency bands (we call this overall average a global threshold $G_{cst}$). The $G_{cst}$ was used as the population mean in a one-sample *t*-test, which tested whether the distances among connectivity patterns of emotion groups were significantly greater than the population mean. The one-sample *t*-test was performed for each frequency band.

Temporal variability of dynamic functional connectivity (tvDFC): The dynamic functional connectivity calculation has been used in the literature during resting state, task conditions, and drug-induced conditions to detect time-varying changes in functional connectivity (FC) measurements [20]. The most commonly used DFC estimation method in the literature is the sliding window approach with fixed-length segments. The network characteristics for each time segment are used to quantify temporal variability in the dynamic functional connectivity [21]. Calculation of tvDFC is a relatively newly developed analysis technique [22,23], which had been used in task-free [24], task-based [25,26], and clinical [27,28] conditions. The temporal variability of different brain regions reflects the dynamic association of these regions in functional modules based on the task demand, promoting the brain’s flexibility and adaptability.

The significantly connected electrode pairs at each time segment were considered to obtain the temporal variability. We followed an approach similar to Rolls et al. [54]. The difference is that they calculated the temporal variability of limited brain regions, whereas we calculated for the whole scalp. Another difference is that instead of using a similarity index such as a correlation coefficient (to achieve dissimilarity, they subtracted the mean correlation coefficient from 1), we used a dissimilarity metric (euclidean distance). The dissimilarity metric is calculated between consecutive time segments. The mean of these dissimilarity metrics was used as the measure of temporal variability.

*Regression analysis between self-assessment scales and temporal variability:* To understand the relation between different dimensions of emotional experience and temporal dynamics of functional connectivity, we performed regression analysis. We fit linear mixed effects models with rating scales as the outcome variable, temporal variability as fixed effects and subject as a random intercept. For the mixed effect regression models, *bobyqa* optimizer was used. The coefficient calculation was made using the REML procedure. Significance and confidence intervals were calculated using the Wald test. All model assumptions were checked for conformity. The regression model was,
$$scale∼tv+\frac{1}{subject}$$
where *scale* outcome variable represents different self-assessment scales, *tv* predictor variable represents the temporal variability of functional connections and *subject* is the random effect.

#### 2.5.5. Hub Calculation

Hubs are regarded as the center of the information flow in the functional networks [55,56]. During the processing of natural audiovisual stimuli, the hub connectivity in the beta band is increased compared to the alpha band [55]. A multimodal study with EEG and fMRI probed the hub dynamics during the negative affective experience and identified the DLPFC as the central hub using IAPS pictures [56].

We calculated hubs on the scalp network to know the hubs’ configuration in the network profile for different emotions. We used eigenvector centrality-based calculation to find hubs. A node in consideration will have a high eigenvector centrality score if it is connected to another node with higher centrality than the peripheral node. Intuitively, the eigenvector centrality score shows that if a node (in consideration) is connected to a hub node than some peripheral nodes, it is processing more information. Hence, eigenvector centrality identifies which node has a wide-reaching influence within a given network [57]. The significance of the centrality values was checked against several random graphs. Ten thousand Erdos-ranyi random models were created with preserved connection probability to perform the significance test. The hubs with less than a 5% chance of being randomly selected were designated as the significant hubs.

### 2.6. Code Accessibility

The code/software described in the paper is freely available online at [58]. The code is available as Extended Data. The code was run on Linux systems. The python-3.8, matlab (R2019b) and R version 3.6.3 were used for the programming.

## 3. Results

### 3.1. Distribution of Labeled Emotional Experiences

The stimuli were taken from a validated dataset [33]. We selected stimuli from the dataset that had primarily elicited 16 emotional categories, including Adventurous, Afraid, Alarmed, Amused, Angry, Aroused, Calm, Disgust, Enthusiastic, Excited, Happy, Joyous, Melancholic, Miserable, Sad, and Triumphant. Although the probability of eliciting a particular category was not very high, we observed that the emotion labeling by participants during the EEG study generally matched with the top two labels reported in the dataset (the distribution of rated emotion categories by our participants is shown in Figure 4). We could think of the primary emotional category as target emotions, but our purpose was not to compare the emotional categories elicited in our study with those elicited in the video stimuli database [35].

### 3.2. Band Specificity for Emotion Processing and Connectivity Networks for Different Emotion Groups

We wanted to see in which frequency band the connectivity profile of emotion groups was more different. The connectivity profiles related to different emotion groups for different segments are shown in Figure 3. EEG functional networks were created by calculating phase-based synchronization metric (PLV) among pairs of EEG electrodes. Only the significant connections (non-parametric permutation test with corrected p<0.01) were considered to constitute the functional networks for different emotion groups. These functional links among pairs of nodes were considered to create functional link vectors. It was a weighted vector with the weight of the link if the link is present in the network, otherwise it is set at zero. The distance among these vectors for different emotion groups was calculated for each segment in each frequency band. We calculated the average distance for each emotional pair by pooling the distances across all the segments in each frequency band. The calculated average distances among all pairs of emotion groups for each frequency band were considered to calculate the global average $G_{cst}$. The $G_{cst}\;=\;8.24$ was used as the population mean in one-sample *t*-test. Only for upper beta and lower beta bands, distances among connectivity patterns (28 pairwise comparisons among eight emotion groups) were significantly greater than $G_{cst}$ (Bonferroni corrected for multiple comparisons; upper beta− t(27) = 3.28, p = 0.008, d = 0.63 (medium); lower beta− t(27) = 2.88, p = 0.023, d = 0.54 (medium). Moreover, we tested which frequency band had relatively more distinct connectivity patterns. We performed a pairwise *t*-test among lower beta and upper beta bands. We found that distances among connectivity patterns in the upper beta band were significantly greater than lower beta (t(27) = −2.23, p = 0.034, 95%CI = [−3.76 −0.16], d = −0.62 (medium)) band. Hence, we did further analysis with the signal in the upper beta frequency range.

### 3.3. Temporal Variability of Emotional Functional Networks Are Correlated with Emotional Arousal and Dominance

We used the segment-wise functional connectivity to calculate temporal variability in functional networks. The calculated temporal variability was checked for the regression with different dimensions of emotional experience. The regression analyses showed that the temporal variability of functional connectivity was significantly correlated with arousal followed by dominance dimensions of emotional experience (respectively, β = 0.514, SE = 0.073, 95%CI = [0.37, 0.66], t(229.56) = 7.03,p<0.001; β = 0.31, SE = 0.078, 95%CI = [0.16, 0.46], t(221.919) = 4,p<0.001). The relationship was stronger for arousal than dominance. Following these results, we further checked whether the interaction between arousal and dominance is the predictor of probability of high temporal variability using logistic mixed effect model. We did not find any such relationship.

### 3.4. Nodes Contributing More in Temporal Dynamics of Emotional Functional Networks

We performed hub analysis to find the influential nodes that contributed more to the temporal variability of the functional connectivity. We identified 15 hub electrodes with the significant centrality values (p<0.05). These hub nodes with centrality values were P9 (0.254), F10 (0.195), Fpz (0.173), Fz (0.215), TP9 (0.206), CP1 (0.156), F2 (0.172), P1 (0.174), T9 (0.248), O2 (0.178), P2 (0.16), TP10 (0.156), TP8 (0.156), FC1 (0.207), and Poz (0.172) (Figure 5). As can be seen in the EEG layout (Figure 5), hubs with high connections and centrality are lateralized to the right frontal region, whereas on the posterior sites, hub electrodes with high centrality and connections are bilateral.

## 4. Discussion

In this study, we used the more naturalistic video stimuli, given that constructivist theories of emotion argue that our emotions depend on context [59]. Video stimuli allowed us to present a narrative context that contributes to elicit different emotions. Such stimuli allow us to better understand the neural mechanisms underlying emotions and also enable us to better generalize to real-life contexts [3]. However, our feelings may change as we watch a video or a movie and elicit multiple emotions [30]. This requires one to find the time at which emotion was experienced to isolate better the neural mechanisms specific to that emotional experience [3]. An important contribution of the study is the temporal localization of emotion experience to probe the frequency band that best captured the brain dynamics of such experiences.

We found that the connectivity patterns in the upper beta band could represent distinct emotion groups better than in other bands. The hubs of the networks were mostly found in the fronto-temporo-parietal (FTP) sites. In a multi-modal EEG-fMRI study, activity in these FTP sites was related to activity in Insula, parahippocampal gyrus (posterior temporal lobes) and ACC [56]. Furthermore, the metric quantifying the temporal variability of the network dynamics is correlated with arousal and dominance subjective ratings.

### 4.1. Emotion-Specific Activity in the Beta Band

We observed significant differences in the connectivity patterns of different emotions in the upper beta band compared to other bands. The higher the difference, the lesser the overlap among connectivity vectors for emotions and the higher the chances of distinct network profiles for different emotion groups. Activity in the upper beta band was observed in some studies using static stimuli such as affective pictures [5,6,12] and emotional faces [60]. For instance, a coherent beta band brain activity in the prefrontal and posterior sites was observed while participants were stimulated with the high arousing IAPS affective pictures [5] and unpleasant and pleasant IAPS pictures [6,12].

Very few EEG studies with multimedia emotional stimuli [19] exist. An earlier study with emotional movies [19] found differences in PLV values in the beta band but also in the alpha and gamma bands. We found connectivity differences only in the beta band. The difference in results could be due to multiple methodological differences between the two studies. We restricted our analysis to segments during or just before the report of the emotional experience instead of the whole stimulus. In addition, they used a shorter duration stimulus and repeated the stimuli in different formats.

In the literature, the beta band has been associated with diverse cognitive functions, including maintenance of information in working memory, motor planning, content-specific modulation, decision making, top-down perceptual processing, long-range communication, and preservation of the current brain state [61]. However, recent evidence suggests a role for the beta wave, particularly in content-specific reactivation and maintenance during endogenous information processing as demanded by the current task [61]. Reactivation of content is needed for the construction of perception. Furthermore, the maintenance of the activated information and integration with the information acquired in the current context is supported by the computational model of cell assemblies [62]. Beta-synchronized cell assemblies are self-sustaining even in the absence of continuing input. After receiving further input, these assemblies create coexisting spiking activity rather than creating competitive spiking activity, which promotes the reactivation and maintenance of the information [61,62]. Hence, we suggest that in our results, the reactivation of emotional episodes in the beta band contributes to the distinct connectivity patterns for different emotion groups.

In addition, the multimedia stimuli with video are more effective [19], situated, rich in context and have an explicit temporal order of events compared to static stimuli [3,4,63]. The potentially rich context effects allow us to interpret the results using constructivist theories of emotions. The reconstructed and maintained information in the beta oscillations primarily serve the endogenous top-down-controlled processing through long-range connections [64] and are involved in making top-down predictions. Hence, we suggest that the activity dynamics in the beta band observed in our results are due to emotion-specific content reactivation and maintenance. This re-activated content modulates the cortical processing via top-down activity.

### 4.2. Temporal Variability in Dynamic Functional Connectivity

With our regression modeling, we observed a positive regression coefficient between arousal and temporal variability in the organization of functional networks. High arousal is associated with higher variability in functional networks. Our results are in line with past studies that observed the re-organization of functional networks associated with arousal. For instance, it has been suggested that the arousal system resets functional brain networks in support of specific behavior suited to the environmental demands [65,66]. An fMRI study along with pupilometry has shown a brain-wide decrease in between-network integration at low relative to high arousal [67]. Studies with caffeine have reported higher temporal variability for high arousal [68]. A graph-theoretic study has shown restructuring of the global network in response to high arousal emotional word stimuli [69]. Our work has uniquely studied the relationship between temporal variability of functional networks for emotional experiences and emotional arousal with a more realistic emotional context.

Dominance refers to an individual’s sense of having an ability to control and influence the situation or event and a dimension associated with emotions [46]. The social situation contains greater variations in the dominance-submissiveness dimension [70]. There is very little research on dominance, and our finding that temporal variability of functional connectivity predicts dominance is a novel and potentially important result for dimensional theories of emotional experience.

In this study, we aimed to work with the information that can be best provided by the EEG. EEG has good time resolution while covering the whole scalp space. Hence, working with the time-frequency information could be more informative. We did not aim to find out specific pathways for different emotion categories. Finding the emotion-specific neural pathways would be appropriate with fMRI alone or fMRI together with EEG using the same paradigm, given the better spatial resolution of fMRI. It is to be noted that a meta-analysis on emotion research hints that there is very little evidence for emotion-specific pathways [71]. Of course, this does not exclude the possibility completely. Using the naturalistic emotional paradigm that can capture emotional events (as we presented in the study) could give some interesting results with better spatial and time resolution recordings of brain activity.

### 4.3. Hub Activity in Right Frontal and Bilateral Posterior Brain Regions

In addition to investigating the temporal variability of dynamic functional connectivity, we also analyzed how the configuration of high information trafficking nodes might reconfigure over time to accommodate the situated emotional experience. Hub calculations were done using eigenvector centrality. Eigenvector centrality has been used in neuroimaging research and reported to be modulated by the current state of the subject [57].

The right hemisphere hypothesis regarding emotions argues for a general dominance of the right hemisphere for all emotions regardless of their valence, especially in the frontal cortex [72,73,74] (although, see [75]). Our hub analysis results show that the hubs in the frontal regions are dominant on the right side and are consistent with the hypothesis that the right frontal cortex plays a significant role in emotional processing. Hubs mediate the information flow in a network and influence the processing of emotional information leading to emotional experience, which seems to be more influenced by right frontal activity. In addition, we observed that this activity takes place in the upper beta band. As described above, the activity in the upper beta band had been reported for the content-based re-activation and maintenance [76,77]. In addition to the maintenance of the task-related content (or the ’status quo’) [61], recent investigations revealed that central and parietal upper beta band activity is related to temporal integration [78]. The temporal integration and the content-specific beta activity may be essential for constructing perception out of a coherent temporal sequence in a naturalistic stimulus. Together, both the functionalities of the upper beta band relate it with the temporal expectations, which may serve the planning of the active and constructive perception of the emotional event.

Recently, a multimodal study using EEG and fMRI was performed with negative and neutral affective pictures as stimuli [56]. They traced the functional connectivity in time and reported four brain networks with their chronological order. Though the timing aspects are not comparable between the two studies (due to the use of static stimuli), the dynamics of brain activity can be compared. In their study [56], the earliest network involves brain regions from the parietal, frontal and occipital lobe (first network); followed by the second network comprising regions in the frontal and temporal lobe; the third network comprising frontal, parietal and occipital; and fourth network occipital, parietal and temporal lobe. In our results (see Appendix A), fronto-parietal-temporal activity in seg-0 is followed by occipital and parieto-occipital regions. In seg-2, activity in parietal and temporo-parietal electrodes is followed by frontal and then frontal, temporal and parietal regions till seg-7. During the seg-8, the activity in frontal, parietal, and temporo-parietal electrodes increased. We witnessed less activity in nodes within the occipital lobe in our results. The primary reason may be that the study [56] was made with affective images, and results are limited to the analysis of 1000 ms only, which might not have captured other aspects of emotions. In comparison, we used emotional film stimuli, and the data analysis was for a longer duration of 7 s.

Our results of changes in the hubs configuration (temporally) are also in line with a recent study [79] probing the temporal flow of hubs for resting state. The non-static and non-re-occurring nature of hub activity is reported to emphasize that there may be no global hub network underlying the well-defined global patterns of shortest paths in brain connectivity. The dynamics of hub configuration adapt to the dynamics of active content representation in the brain.

Groups-2, 3, and 4 show more connectivity both when looking at hubs only (Appendix A) and overall connectivity (Figure 3 and Appendix A). Another pattern that can be observed is that the frontoparietal connectivity is higher in these emotion groups than the emotion groups 1, 5, 6, 7, and 8. These results can be interpreted in light of a study [18] in which static and dynamic conditions for emotional feelings were experimentally created. They found that during the static condition (i.e., no emotion transition), the functional connectivity between prefrontal and parietal scalp electrodes was higher in the beta band, contrary to the dynamic condition (with emotion transition), which showed lower functional connectivity. The rationale behind the decreased functional connectivity is that a more loose prefrontal–posterior coupling may be related to the loosening of the prefrontal cortex’s control over emotional information. Thus, the brain becomes more affected by emotional fluctuations. On the other hand, increased prefrontal–posterior coupling may be related to strong control and the tendency to protect oneself from becoming emotionally affected. Some other studies [80] also reported right frontal and bilateral parietal activity (as we observed in our results in Figure 5 and Appendix A). For instance, judging the emotional context in IAPS stimuli reportedly evoked activity in the right frontal and left parietal regions. In a recent study, beta frequency bands in the right frontal and bilateral parietal lobe were reported to be sensitive to different emotions [81].

The activity in frontal and frontocentral electrodes is reported mainly during interoceptive processing. Interoceptive awareness about the afferent signal from the body contributes in perceiving and feelings of emotional experiences. For instance, [82] reported activity in frontal and frontocentral electrodes during the heartbeat perception task. Our recent study also reported activity in frontal and frontocentral electodes while probing the interaction between cardiac and brain activity [45]. Another study reported activity in frontal electrodes during the emotional processing of IAPS pictures [83,84]. The conscious perception of visceral activity is generally reported in the right hemispheric part of the Insula [85] that contributes in the experience of emotions in the beta band [86]. Marshall et al. [84] reported activity in frontal, frontocentral and frontotemporal electrodes in top-down anticipation of the heartbeat signal with increased sensory perception in the context of emotionally negative and neutral faces. Activity in the medial prefrontal regions of the brain in the beta band showed distinguishing patterns for two positive emotions (tenderness and amusement) [87]. In our case, the electrodes in the frontal midline were active as hubs for both the positive and negative emotions. Activity in the posterior electrodes is reported in the hedonic evaluation by van Bochove et al. [88]. During episodic events retrieval, the functional connectivity between frontal and parietal regions is reported. EEG activity in the right parietal regions in the beta frequency domain was related with the anticipation of avoidant response to angry facial expression [89]. The likely possibility is that the episodic memories related to an emotional event are fetched, which then could motivate the sensory-motor response to an aversive situation.

In terms of applications, the beta band-specific results may be useful in emotion recognition and affective computing. Clinically, studies of schizophrenia often report that the synchronization activity and long-range temporal correlations in the beta band are modified due to clinical conditions [90]. However, it is difficult to directly compare our results with the clinical population because our study is done with a healthy population. However, activity in the beta band during emotional experience could be considered in future research to discover any pathological markers related to emotional experience.

### 4.4. Limitations

One limitation is the narrow age range of the sample population since the sample used in the study are university students. Given that emotional processing changes with age [91], studying dynamic functional connectivity associated with emotions across different age groups would be important. Another limitation is the gender of the sample, which consists of predominantly male students since the sample came from a technological university that consists of a male majority population. Given gender differences in emotional processing [92], there is a need to perform a similar study with male and female participants.

## 5. Conclusions

Our study shows that the connectivity patterns for different emotion groups in the beta band is more distinct than in other bands. Our results show that the pre-experience activity in the brain has enough information in connectivity patterns of different emotions. These connectivity patterns vary with time and are linked to self-assessed dimensions of an emotional experience such as arousal and dominance. The current study did not aim to study the dynamics associated with specific emotions over time, but future studies could focus on specific emotions and their underlying neural mechanisms, including dynamic functional connectivity.

## 6. Data Statement

EEG Dataset is available at openneuro [93].

## Figures and Tables

**Figure 1 brainsci-12-01106-f001:**
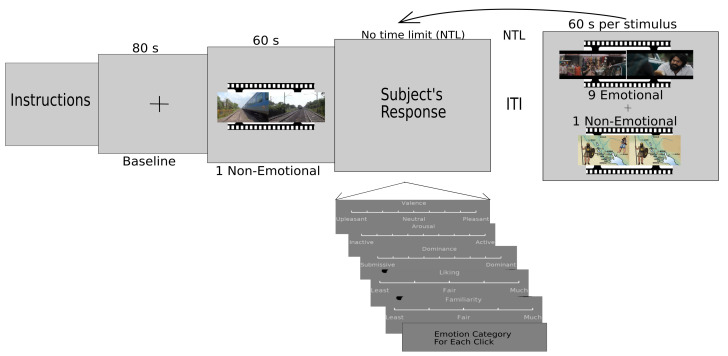
Experiment Paradigm: Each participant was shown 11 60-s long video stimuli. One non-emotional stimulus was shown after the baseline recording. Nine emotional stimuli were presented with order randomized. Another non-emotional stimuli was shown after the fifth and before the eighth video. There was no time limit during the ratings and inter-trial interval. Participants could resume watching the next stimulus after clicking the left button on the mouse.

**Figure 2 brainsci-12-01106-f002:**
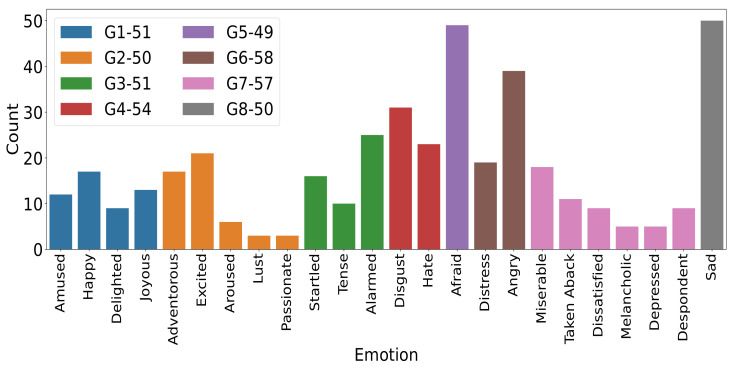
Representation and statistics of emotion groups: The histogram plot shows the number of time participants has rated any emotion category to express their emotional experience. Histogram bars in the same color form an emotion group. The legend shows the number of emotional instances per emotion group. The grouping criteria is described in Section 2.5.1. G1: happy, amused, delighted, joyous; G2: aroused, adventurous, excited, passionate, lust; G3: startled, tense, alarmed; G4: disgust, hate; G5: afraid; G6: distress, angry; G7: miserable, taken aback, dissatisfied, melancholic, depressed, despondent; G8: sad.

**Figure 3 brainsci-12-01106-f003:**
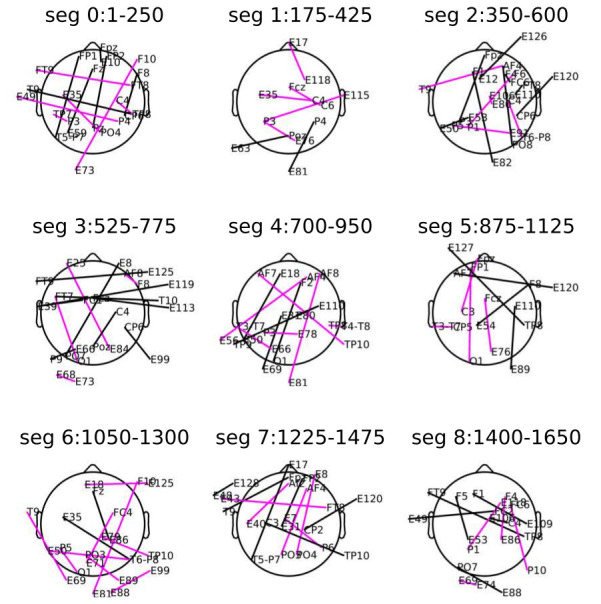
**Significant functional connections in the upper beta band:** Only significant connections are plotted. The color coding is as per the stats value as follows: magenta (5<stats≤6), black (6>stats). Above each plot the information about duration of the segment is provided. Functional connections for Group-03 is presented here. See Appendix A for plots related to other groups.

**Figure 4 brainsci-12-01106-f004:**
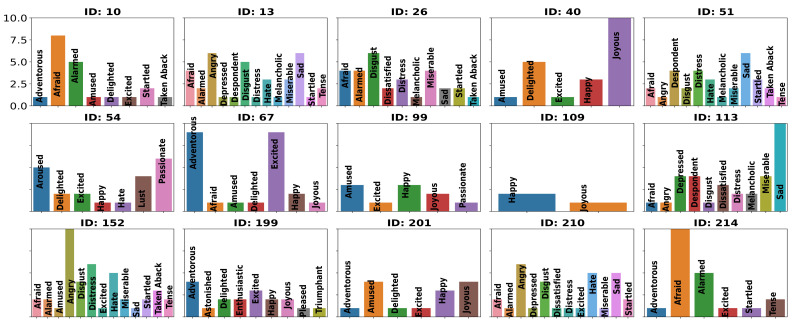
The figure shows the distribution of emotion categories in which participants rated their emotional events while watching the stimulus. The title of each subgraph shows stimulus ID. The number of times participants rated an emotional category is shown on the y-axis.

**Figure 5 brainsci-12-01106-f005:**
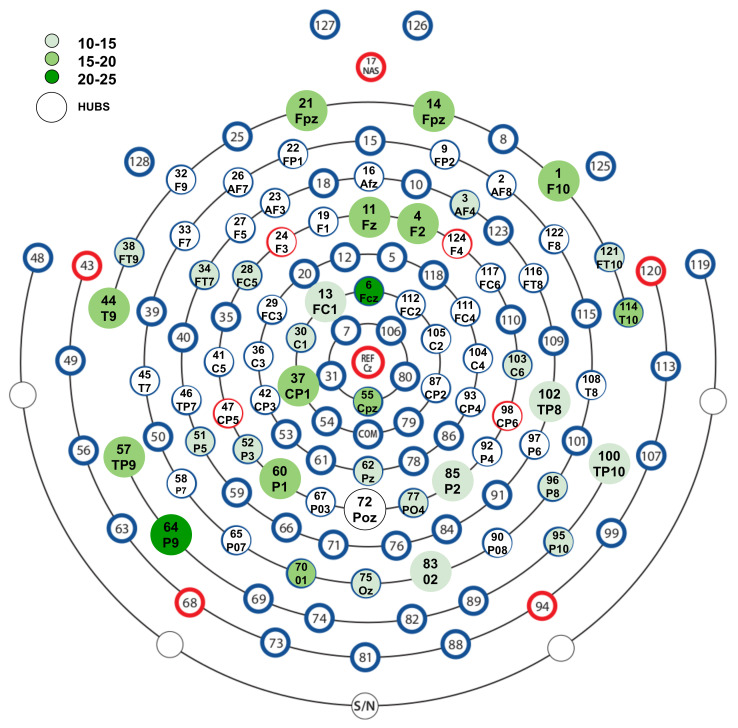
**Representation of hubs on the layout of used EEG cap:** Map of EEG electrodes with color code according to the number of times the node is active in the network across emotion groups. The electrodes with a large size are calculated as hubs using eigenvector centrality with less than 5% chance of being falsely detected as hubs (permutation test with 10,000 Erdos-ranyi random models). See Appendix A, Appendix A. Abbr. are– Fc: fronto-central, Fp: Frontopolar, Af: Anterior frontal, CP: centroparietal, TP: temporo-parietal, PO: parietooccipital.

## Data Availability

The data presented in this study are openly available in openneuro https://doi.org/10.18112/openneuro.ds003751.v1.0.3 (accessed on 13 April 2022).

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
