# Peer review of "Dynamic Functional Connectivity of Emotion Processing in Beta Band with Naturalistic Emotion Stimuli"

_brainsci, 2022, doi:10.3390/brainsci12081106_

Round 1

Reviewer 1 Report

I have reviewed the manuscript titled “Dynamic functional connectivity of emotion processing in beta band with naturalistic emotion stimuli”. The overall contents of this manuscript are not well organized to give a clear overview of this work. I have suggested some comments about this work are as the following:

Comments to the Authors:

1.     Authors should write abstract clearly including, background, method, results, significance.

2.     In method section author should add one more figure related to the system architecture diagram of emotions/flowchart for experiment design and EEG signal process.

3.     Explain type of emotion like, happy, sad, joy, surprise, sadness, anger, disgust, fear, shame, shyness, and guilt.

4.     In result section, author should perform time and frequency analysis at 1-50 Hz frequency.

5.     In results section, author should clearly highlight the neural network pathways of emotions/functional connectivity network of emotions.  

6.      My suggestion is that the authors should write discussion section clearly in more details like how and why this functional study is important than previous clinical research.

7.     The authors should write some limitations of this study and application in more details.

Reviewer 2 Report

The paper was well-written with a detailed introduction that covers the topic well.

The methods, although understanderbale, could benefit fromm some figures that give detail to the flow of the experimental procecudure- what happed when and what was taken (EEG, self-assessment). It as difficult to follow from the text alone. For instance the first self-rating (lines 159-162) as this to obtain the baseline rating before the videos?

Other methods queries I have about the methods are:

-Why were 20 participants  (line 130) shown stimuli when there were 40 participants?

 -The ages band was rather young (average 23 years) it would be better if a more normative group was investigated.

-Give a reference for the 30KOhm impedance- as this is rather high and may also explain why there were very high amplitudes from detached electrodes.

-Lines 163 to 172 are background information that trys to explain important concepts but reads too much like background rather than methods- Try and rewrite this so that there is relevance to the methods.

In the results  (3.2) what was upper and lower beta the only comparison? 

Figure 3- Why was there only 15 particpants?

-Any limitations that need to be discussed?

Reviewer 3 Report

Comments/Suggestions for Authors

In this study, Mishra and colleagues investigated the dynamic functional connectivity of emotion processing during naturalistic emotion stimuli of young subjects. Significant differences were observed in the upper beta frequency band. They showed that the connectivity patterns in the upper beta band could differentiate emotion categories better during or prior to the reported emotional experience. Furthermore, they showed that the functional connectivity dynamics were primarily related to emotional arousal followed by dominance, and the hubs in the functional networks were found across the right frontal and bilateral parietal lobes. This is an interesting study, which might contribute to a better understanding of dynamic emotional processing. I have no major concerns and specific comments/questions are given below.

Introduction

P. 1, L. 32-33: Something seems wrong here. The ERP analysis and Beta band analysis (probably EROs analysis) are different analyses. Therefore, the term ‘‘ERP amplitude in beta band’’ is wrong. In addition, I think that the reference is incorrect in this sentence. I would recommend checking this sentence and reference.

P. 1, L. 37-38: What are other frequencies which are mentioned in this sentence? Briefly, other frequencies can be mentioned in one or two sentences. In addition, the literature another than those cited in the text can be added.

P. 3, L. 117-120: The hypothesis of the study should be improved.

Materials and Methods

P. 3, L. 122-126: It is known that gender effect EEG responses in emotion studies. Especially, gender differences influence the brain's beta responses to emotion. Were only 3 females included in the study? If so, the gender effect should be eliminated from the study (new female participants should include in the study) or this should be added as a limitation of the study.

What is the duration of stimuli (videos) and inter-stimulus interval?  It should be indicated stimuli (videos) duration and inter-stimulus interval in the section of the experimental paradigm.

To be more descriptive, it could be added a figure of the experimental paradigm.

P. 4, L. 188: Which were used electrodes for EEG recording? Active or passive electrodes? In both cases, the impedance value of the electrodes is very high (below 30 Kohm). I would recommend adding this as a limitation of the study.

In Figure 1, words could be presented in the smaller point size (emotion groups: amused, happy, delighted, etc.)

Results

No comment

Discussion

P. 10, L. 401-402 and P. 12, L. 403: It could be added fMRI studies which indicated the relation between emotion and cingulate cortex, parahippocampal gyrus, and insula.

P. 12, L. 445-453: These sentences are like a repetition of the sentences in the upper paragraph (Line 425-433).

Round 2

Reviewer 2 Report

The authors have addressed all my concerns. 

Reviewer 3 Report

The authors have addressed my remarks adequately, and I have no other comments.